# Prediction Horizon-Varying Model Predictive Control (MPC) for Autonomous Vehicle Control

Zhenbin Chen, Jiaqin Lai, Peixin Li *, Omar I. Awad and Yubing Zhu

School of Mechanics and Electronics Engineering, Hainan University, Haikou 570228, China;
990516@hainanu.edu.cn (Z.C.); 1330lai@hainanu.edu.cn (J.L.); 184424@hainanu.edu.cn (O.I.A.);
zhuyubing@hainanu.edu.cn (Y.Z.)
* Correspondence: 993908@hainanu.edu.cn

**Abstract:** The prediction horizon is a key parameter in model predictive control (MPC), which is related to the effectiveness and stability of model predictive control. In vehicle control, the selection of a prediction horizon is influenced by factors such as speed, path curvature, and target point density. To accommodate varying conditions such as road curvature and vehicle speed, we proposed a control strategy using the proximal policy optimization (PPO) algorithm to adjust the prediction horizon, enabling MPC to achieve optimal performance, and called it PPO-MPC. We established a state space related to the path information and vehicle state, regarded the prediction horizon as actions, and designed a reward function to optimize the policy and value function. We conducted simulation verifications at various speeds and compared them with an MPC with fixed prediction horizons. The simulation demonstrates that the PPO-MPC proposed in this article exhibits strong adaptability and trajectory tracking capability.

**Keywords:** trajectory tracking; MPC; prediction horizon; PPO reinforcement

## 1. Introduction

Nowadays, advancements in autonomous driving technology have led to significant enhancements in road safety and driving efficiency [1–3]. As a pivotal technology, the effectiveness of trajectory tracking control directly affects the overall performance and safety of vehicles during operation [4]. Despite significant progress in autonomous driving technology, optimizing trajectory tracking control remains a challenging but crucial task [5]. Moreover, as the demand for autonomous vehicles continues to increase, it is crucial to refine trajectory tracking control algorithms to effectively address complex and various road scenarios [6,7], achieving a good balance between responsiveness, accuracy, and computational efficiency in dynamic driving environments and ensuring real-time adaptability.

One main method to solve trajectory tracking control problems is model predictive control (MPC) [7,8]. MPC is a method for solving optimal control problems (OCPs) using the current state of the controlled system as the initial condition. This method uses a controlled object model to predict the controlled variable's response [9,10]. Solving an OCP involves finding a sequence of control inputs that minimize the objective function within the specified prediction horizons. Simultaneously, it maintains feasibility while the trajectory stays within the defined limitations. The linear time-varying model predictive control (LTV-MPC) proposed by [11,12] is an MPC based on the discrete linear state space model, which can reduce the amount of calculation and improve efficiency. The LPV-MPC presented in [13] takes into account future inputs and scheduling parameters, predicting future outputs accordingly. Ref. [14] presents a customized genetic algorithm for real-time optimization of a nonlinear model predictive control (NMPC) path-tracking controller, specifically designed for lower vehicle speeds.

With increasing computing power, sensing, and communication capabilities and advances in the field of machine learning, automating controller design and adaptation

based on data collected during operations are being studied [15], such as by improving performance, facilitating deployment, and reducing the need for manual controller tuning. In refs. [16–18], Gaussian Process Regression and neural networks were used as predictive models of controlled systems, which were prediction models adaptively adjusted in a data-driven manner to improve control accuracy and reduce computing costs. The concept of using reinforcement learning (RL) to learn MPC cost function parameters is introduced in ref. [19]. Ref. [20] proposes a weights-varying MPC using a deep reinforcement learning (DRL) algorithm to adjust cost function weights in different situations. Ref. [21] proposed a novel approach limiting DRL actions within a safe learning space, and the proposed DRL algorithm can automatically learn context-dependent optimal parameter sets and dynamically adapt for a weights-varying MPC. Ref. [22] has introduced a novel control algorithm based around an event-triggered MPC, using RL with a configurable objective to automatically tune the control algorithm's meta-parameters: the prediction horizon and the re-computation (triggering) problem. In ref. [23], a RLMPC scheme was introduced integrating MPC and RL through policy iteration (PI), where MPC is a policy generator and the RL technique is employed to evaluate the policy. This RLMPC scheme has great potential in reducing computational burden.

Prediction horizon is the key parameter affecting both performance and computational burden of the control system in MPC. It denotes the time range used to predict the system's future behavior during the control process [24]. Shorter horizons offer better control but less stability, while longer horizons provide stability but should not be excessively long [25]. In ref. [26], a dual-mode receding horizon controller eliminates the danger of interference that is always present in nonlinear optimal control algorithms and greatly reduces the amount of online calculation required. Two adaptive prediction horizons for MPC were proposed in ref. [27]. One is based on heuristics, which is idealized but not feasible, and the other is more practical and uses iterative deepening, where each iteration will check stability criteria and find out the minimum horizon of stability. Ref. [28] adjusted the prediction horizon range to a positive integral discrete time variable and introduced an NMPC to achieve velocity control, incorporating a self-correction method for the prediction horizons. Studies in refs. [29,30] proposed an event-triggered MPC, which dynamically adjusts the prediction horizon using event-triggering mechanisms to facilitate the optimization process. A finite control set MPC with an adaptive prediction horizon is proposed [31], and neural networks are trained to calculate the optimal prediction horizon at runtime. Ref. [9] proposed an RLMPC which learns the optimal prediction horizon length of an MPC scheme using RL.

Proximal policy optimization (PPO) is a policy-based RL algorithm, clipping the probability ratio to modify the agent's objective and constraining the magnitude of policy change at each step to enhance training stability. In this paper, we propose an adaptive MPC, combining the PPO algorithm and MPC to adjust the prediction horizon. To accommodate dynamic conditions such as road curvature and vehicle speed, we leverage the PPO algorithm to adaptively adjust the prediction horizon within the MPC framework. This integration empowers MPC to achieve optimal performance under varying environmental circumstances.

Unlike other research that combines reinforcement learning with MPC for autonomous driving—such as using RL for planning, MPC for control, and RL to adjust MPC weight parameters—our proposed PPO-MPC deeply considers the impact of prediction horizon selection on control performance, offering a dynamic prediction horizon that enhances MPC's adaptability. And the PPO-MPC strategy we propose enables the vehicle to achieve adaptive tracking control at different speeds and curvatures.

## 2. Methodology

This section outlines the methodology of our proposed PPO-MPC framework, encompassing the establishment of the vehicle dynamics model, the design of the MPC strategy, and the adaptation of prediction horizons within the PPO algorithm. Specifically, we introduce a novel approach called "prediction horizon-varying model predictive control" to optimize the prediction horizon for MPC. This involves formulating a hybrid PPO-MPC prediction horizon optimization problem. To improve the adaptive performance of autonomous driving trajectory tracking control, we use the PPO algorithm to dynamically adjust the MPC prediction horizon, exploring the intricate relationship between the prediction horizon and variables such as vehicle speed, lateral deviation, and curvature. A visual representation of the architectural concept of PPO-MPC is provided in Figure 1. Figure 1 shows that the vehicle dynamics model is used to analyze the vehicle's motion state; the lateral error model and the longitudinal acceleration model are coupled, with the front wheel steering angle and acceleration serving as control variables. And the model predictive control algorithm is used to solve the problem. In this structure, a state space related to the MPC controller and vehicle motion status is established, and the prediction horizon is set as the action space. Based on the PPO algorithm, the optimal prediction horizon is dynamically adjusted through iterative training. In addition, the speed control strategy can be interpreted as calculating the desired acceleration through model predictive control, switching the driving mode through the desired acceleration corresponding control strategy and then controlling the vehicle's accelerator opening and brake pressure to achieve speed control.

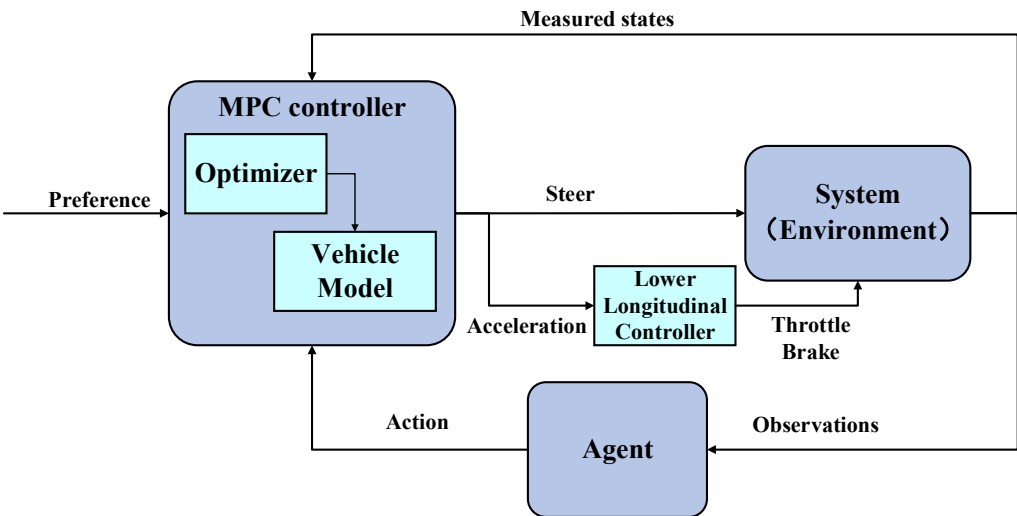

**Figure 1.** The concept architecture.

### 2.1. Vehicle Dynamic Model

As shown in Figure 2, this study considers the vehicle dynamics model of both longitudinal and lateral motion.

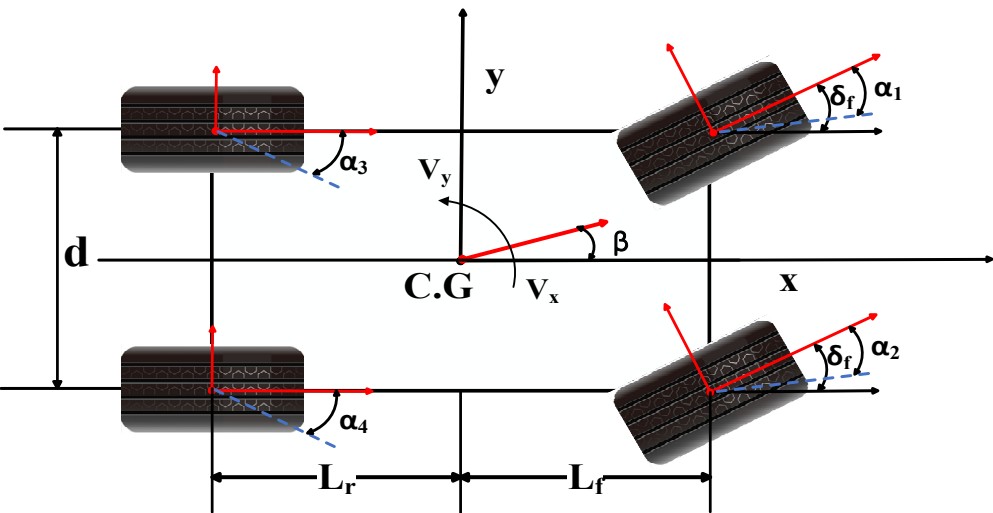

**Figure 2.** Vehicle dynamics model.

2.1.1. Lateral Dynamics

In this paper, the model for lateral movement is constructed on the foundational principles of the bicycle model. This approach presumes symmetry in the steering angles on both the left and right sides, which effectively reduces the complexity of the vehicle's dynamic model to that of a two-wheeled bicycle. Leveraging Newton's second law of motion, simplified Equations (1a)–(1c) can be derived to quantify the lateral force exerted by the vehicle.

$$m\ddot{y} = m\dot{x}\dot{\varphi} + 2F_{xf} + 2F_{xr} \tag{1a}$$

$$m\ddot{x} = m\dot{y}\dot{\varphi} + 2F_{yf} + 2F_{yr} \tag{1b}$$

$$I_z\ddot{\varphi} = 2l_f F_{yf} - 2l_r F_{yr} \tag{1c}$$

where $m$ is the vehicle mass and $\dot{x}$ and $\dot{y}$ are the longitudinal and lateral velocity, respectively. $\ddot{x}$ and $\ddot{y}$ denote the longitudinal and lateral acceleration, respectively. $F_{yf}$ and $F_{yr}$ are the lateral tire forces at the front and the rear wheels, respectively; $F_{xf}$ and $F_{xr}$ are the longitudinal tire forces at the front and the rear wheels, respectively; $\dot{\varphi}$ is the yaw rate; $I_z$ denotes the yaw inertia of the vehicle; and $l_f$ and $l_r$ are the distances from the front and rear axles to the center of gravity, respectively.

Considering the tire turning characteristics, Equations (2a) and (2b) represent the lateral force of the front and rear tires at a small sideslip angle:

$$F_{yf} = 2C_f a_f \tag{2a}$$

$$F_{yr} = 2C_r a_r \tag{2b}$$

where $C_f$ and $C_r$ are the cornering stiffness of the front tire and rear tire, respectively; $\alpha_f$ is the sideslip angle of the front tire; and $\alpha_r$ is the sideslip angle of the rear tire.

Given that the angles of the two front wheels of the vehicle are equal, the vehicle's lateral acceleration should meet the criteria for the small angle assumption. In this case, the subsequent approximate relationship can be utilized:

$$a_f = \delta_f - \frac{l_f\dot{\varphi} + \dot{y}}{\dot{x}} \tag{3a}$$

$$a_r = \frac{b\dot{\varphi} - \dot{y}}{\dot{x}} \tag{3b}$$

Upon substituting Equations (2a), (2b) and Equations (3a), (3b) into Equations (1a)–(1c), the resulting expressions are obtained:

$$m\ddot{y} = -mV_x\dot{\varphi} + 2\left[C_{cf}(\delta_f - \frac{\dot{y} + l_f\dot{\varphi}}{\dot{x}}) + C_{cr}\frac{l_r\dot{\varphi} - \dot{y}}{\dot{x}}\right] \tag{4a}$$

$$m\ddot{x} = m\dot{y}\dot{\varphi} + 2\left[C_{cf}(\delta_f - \frac{\dot{y} + l_f\dot{\varphi}}{\dot{x}}) + C_{lf}S_f + C_{lr}S_f\right] \tag{4b}$$

$$I_z\ddot{\varphi} = 2\left[l_fC_{cf}(d_f - \frac{\dot{y} + l_f\dot{\varphi}}{\dot{x}}) - l_rC_{cr}\frac{l_r\dot{\varphi} - \dot{y}}{\dot{x}}\right] \tag{4c}$$

### 2.1.2. Longitudinal Dynamics

Vehicle longitudinal dynamics studies the motion and mechanical characteristics of a vehicle in the longitudinal direction (i.e., the direction of the vehicle's forward motion). In longitudinal dynamics, the vehicle's acceleration, braking, traction, resistance, and other factors are primarily considered. Newton's second law provides a framework for understanding these forces and their impact on motion. Considering the computational complexity, the effect of partial resistance is not taken into account.

$$m\ddot{x} = F_t - F_w - R_x \tag{5}$$

where $F_t$, $F_w$, and $R_x$ respectively represent driving force, air resistance, and rolling resistance. The expression for air resistance $F_w$ is as follows.

$$F_w = \frac{1}{2}\rho_{\text{air}}C_dA_f{v_x}^2 \tag{6}$$

where $\rho_{\text{air}}$ represents air density. $C_d$ stands for the drag coefficient. $A_f$ denotes the frontal area of the vehicle.

Torque is based on the sum of the forces multiplied by the wheel radius; according to Gillespie's "Fundamentals of Vehicle Dynamics" [32], we can obtain the calculation of torque in the context of vehicle dynamics, denoted as Equation (7).

$$T_t = r_{tire}(m\ddot{x} + \frac{1}{2}\rho_{\text{air}}C_dA_f{v_x}^2 + R_x) \tag{7}$$

where $T_t$ is the driving torque and $r_{tire}$ represents the effective tire radius.

Taking into account the transmission ratio and motor efficiency, the relationship between driving force and motor torque is:

$$T_m = \frac{r_{tire}(m\ddot{x} + \frac{1}{2}\rho_{\text{air}}C_dA_f{v_x}^2 + R_x)}{i_0\eta_t} \tag{8}$$

where $T_m$ denotes the motor torque, $i_0$ is the transmission ratio, and $\eta_t$ is the motor efficiency. The braking force can be expressed as:

$$F_d = -ma_{des} - \frac{1}{2}\rho_{air}C_dA_f v_x^2 - R_x \tag{9}$$

Considering that the braking and driving modes cannot work at the same time, and the braking and driving modes cannot be switched frequently, the driving and braking switching strategy of the vehicle is designed as:

$$\text{mode} = \begin{cases} 1(\text{driving}), a_{\text{des}} \geqslant a_{\text{thre}} + 0.1\,\text{m/s}^2; \\ 0(\text{nochange}), a_{\text{thre}} - 0.1\,\text{m/s}^2 < a_{\text{des}} < a_{\text{thre}} + 0.1\,\text{m/s}^2; \\ -1(\text{braking}), a_{\text{des}} \leqslant a_{\text{thre}} - 0.1\,\text{m/s}^2 \end{cases} \tag{10}$$

### 2.2. MPC System Definition

MPC is a control system approach employing a predictive model. In each discrete sampling time, MPC solves an open-loop OCP over a predetermined finite horizon. This is an iterative process of predicting the future system behavior within the future horizon through the plant's predictive model. The current state of the system is considered the initial condition for each optimization cycle. An optimizer solves the optimization problem and returns a control sequence for the prediction horizon. The plant only adopts the initial control in the optimal sequence, and on subsequent samples, the initial control is used to solve the system optimization problem again [33]. This refers to the "receding horizon" principle [34]. Indeed, the horizon recedes as time passes. A key feature of MPC is its ability to integrate hard constraints on control variables and states during the design phase.

In this study, we divide longitudinal control into two parts: upper-level and lower-level control. Upper-level control uses MPC, producing acceleration as its output. Lower-level control is deduced using Equation (7), which establishes a throttle and brake control map.

According to Ref. [35], the longitudinal accelerating system is modeled as a linear first-order system; the relationship between the desired vehicle acceleration $a_{des}$ and the actual acceleration is as follows in Equation (11).

$$\ddot{x} = \frac{K}{\tau s + 1} a_{des} \tag{11}$$

where $K = 1$ is the gain coefficient, $\tau$ is the delay time, and $\tau = 0.5$.

The longitudinal accelerating and lateral steering combined system model of the vehicle can be described as Equation (12).

$$\frac{d}{dt}\begin{bmatrix} \ddot{x} \\ \dot{x} \\ y \\ \dot{y} \\ \varphi \\ \dot{\varphi} \end{bmatrix} = \begin{bmatrix} -\frac{1}{\tau} & 0 & 0 & 0 & 0 & 0 \\ 1 & 0 & 0 & 0 & 0 & 0 \\ 0 & 0 & 0 & 1 & 0 & 0 \\ 0 & 0 & 0 & -\frac{2C_f+2C_r}{m\dot{x}} & 0 & -\dot{x}-\frac{2l_fC_f-2l_rC_r}{m\dot{x}} \\ 0 & 0 & 0 & 0 & 0 & 1 \\ 0 & 0 & 0 & -\frac{2l_fC_f-2l_rC_r}{I_z\dot{x}} & 0 & -\frac{2l_f{}^2C_f+2l_r{}^2C_r}{I_z\dot{x}} \end{bmatrix}\begin{bmatrix} \ddot{x} \\ \dot{x} \\ y \\ \dot{y} \\ \varphi \\ \dot{\varphi} \end{bmatrix} + \begin{bmatrix} \frac{1}{\tau} & 0 \\ 0 & 0 \\ 0 & 0 \\ 0 & \frac{2C_f}{m} \\ 0 & 0 \\ 0 & \frac{2l_fC_f}{I_z} \end{bmatrix}\begin{bmatrix} a_{des} \\ \delta_f \end{bmatrix} \tag{12}$$

The full system model consists of the longitudinal and lateral model. Equation (13) represents the system's state space expression,

$$\begin{cases} \dot{\xi} = A\xi + Bu \\ y(\xi) = C\xi \end{cases} \tag{13}$$

where $A = \begin{bmatrix} -\frac{1}{\tau} & 0 & 0 & 0 & 0 & 0 \\ 1 & 0 & 0 & 0 & 0 & 0 \\ 0 & 0 & 0 & 1 & 0 & 0 \\ 0 & 0 & 0 & -\frac{2C_f+2C_r}{m\dot{x}} & 0 & -\dot{x}-\frac{2l_fC_f-2l_rC_r}{m\dot{x}} \\ 0 & 0 & 0 & 0 & 0 & 1 \\ 0 & 0 & 0 & -\frac{2l_fC_f-2l_rC_r}{I_z\dot{x}} & 0 & -\frac{2l_f{}^2C_f+2l_r{}^2C_r}{I_z\dot{x}} \end{bmatrix}$, $B = \begin{bmatrix} \frac{1}{\tau} & 0 \\ 0 & 0 \\ 0 & 0 \\ 0 & \frac{2C_f}{m} \\ 0 & 0 \\ 0 & \frac{2l_fC_f}{I_z} \end{bmatrix}$,

$C = \begin{bmatrix} 0 & 1 & 0 & 0 & 0 & 0 \\ 0 & 0 & 0 & 0 & 0 & 0 \\ 0 & 0 & 1 & 0 & 0 & 0 \\ 0 & 0 & 0 & 1 & 0 & 0 \\ 0 & 0 & 0 & 0 & 1 & 0 \\ 0 & 0 & 0 & 0 & 0 & 1 \end{bmatrix}$.

Where $\xi = \begin{bmatrix} \ddot{x} & \dot{x} & e_1 & \dot{e}_1 & e_2 & \dot{e}_2 \end{bmatrix}^T$ is the state vector and $u = [a_{des}, \delta]^T$ is the control input vector of the vehicle model.

Equation (13) can be linearized and discretized as follows:

$$\begin{cases} x(k+1) = (I + T_1 A)x(k) + T_1 B u(k) \\ \qquad\quad h(k) = Cx(k) \end{cases} \tag{14}$$

where $T_1$ denotes the sample time; $I$ is the identity matrix; and $A$, $B$, and $C$ are matrices of coefficients.

Based on the known vehicle model and the deviation between the current measurement value and the expected value, the MPC controller predicts the output of the system within $N_p$. By solving the objective function and optimizing the output, a control quantity array in $N_c$ is obtained, and control elements in the first-time interval $\Delta t$ are used as the output quantity. This process is repeated to achieve vehicle tracking along the desired trajectory.

According to the control requirements, the basic principle of MPC is to minimize the performance evaluation function while satisfying the control constraints. Equations (15a)–(15c) represent the objective function.

$$\min_{\Delta u(k)} J(\eta(k), \Delta u(k)) \tag{15a}$$

$$J(\eta(k), u(k)) = J_1 + J_2 + \rho\varepsilon^2 \tag{15b}$$

$$J_1 = \sum_{i=1}^{N_p} \|\eta(k+i \mid k) - \eta_{ref}(k+i)\|_Q^2, \; J_2 = \sum_{i=0}^{N_c-1} \|\Delta u(k+i \mid k)\|_R^2 \tag{15c}$$

where $J_1$ reflects the system's ability to track reference trajectory within prediction horizon $N_p$. This reflects the system's requirement for a smooth change in the control increment within control horizon $N_c$. $Q$ and $R$ are the weight matrices, $\rho$ is the weight coefficient, $\varepsilon$ is the relaxation factor, and $\Delta u$ is the control input increment.

Considering safety constraints and vehicle actuator constraints, the constraint conditions can be expressed as Equation (16).

$$s.t. \begin{cases} u_{\min} \leq u(k+i|k) \leq u_{\max}, i = 0, 1, \cdots, N_c - 1 \\ \Delta u_{\min} \leq \Delta u(k+i|k) \leq \Delta u_{\max}, i = 0, 1, \cdots, N_c - 1 \\ \eta_{\min} \leq \eta(k+i|k) \leq \eta_{\max}, i = 1, 2, \cdots, N_p \end{cases} \tag{16}$$

*2.3. PPO Horizon Policy*

2.3.1. Proximal Policy Optimization (PPO)

The PPO algorithm is an online policy gradient RL algorithm which can deal with problems in continuous state–action spaces. This DRL algorithm learns optimal strategies in interaction with the environment and uses stochastic gradient ascension to optimize the agent objective function [36]. To achieve high cumulative rewards, the PPO algorithm seeks optimal decisions in complex environments by constructing and optimizing policies, generally represented by neural networks. The policy, a parameterized function, maps states to probability distributions of actions. To improve the policy, PPO uses proximal policy optimization and maximizes the objective function. The objective function comprises a loss term for current policy updates and a KL divergence term, ensuring stability by controlling the magnitude of policy updates.

In PPO, the problem is represented as a Markov Decision Process (MDP), which is defined as $(S, A, T_2, R, \gamma)$. $S$ represents the state space, which encompasses the complete set of potential states that the environment can occupy. And $A$ represents the action space, which contains a collection of all possible actions. $T_2$ represents the state transition function, which defines the probability distribution of the environment transitioning to the next state. $R$ is the reward function, which is specified in the given state and represents an immediate reward. $\gamma$ is the discount factor, which determines the importance of future rewards. The pseudocode of PPO is shown in Algorithm 1.

The goal of PPO is to maximize the anticipated total reward while limiting changes in the policy during each update. In this study, Equation (17) is the objective function expression of PPO.

$$L^{PPO}(\theta) = \mathbb{E}\big[\min\big(r_{t_2}(\theta) \cdot \hat{A}_{t_2}, clip(r_{t_2}(\theta), 1 - \epsilon, 1 + \epsilon) \cdot \hat{A}_{t_2}\big)\big] \tag{17}$$

where $r_{t_2}(\theta)$ denotes the ratio of the new strategy to the old strategy and $\hat{A}_t$ indicates the advantage function.

---

**Algorithm 1** PPO algorithm

---

For iteration $i = 1 : N$ do
    For episode $j = 1 : M$ do
        Initialize the weight parameters of the policy network (actor) and value function network (critic). The initial state is determined by the discount factor $\gamma$ and the greedy factor $g$;
        Collect experiences $D = \{(s, a, r, s') \cdots\}$;
        For optimization step $k = 1 : K$ do
            Calculate the expected advantage function of the current strategy $A(s, a; \theta)$;
            Calculate the advantage function $A(s, a; \theta) - r$ for each experience $(s, a, r, s')$;
            Update the policy $\theta$;
            Update proximal policy $\pi_\theta$;
        End
    End

---

End for

---

The framework of PPO-MPC is shown in Figure 3, which includes the environment and PPO network. The state given in the environment is input into the PPO network for learning and training. The state quantity is scored through the critic network, and then the appropriate action is selected.

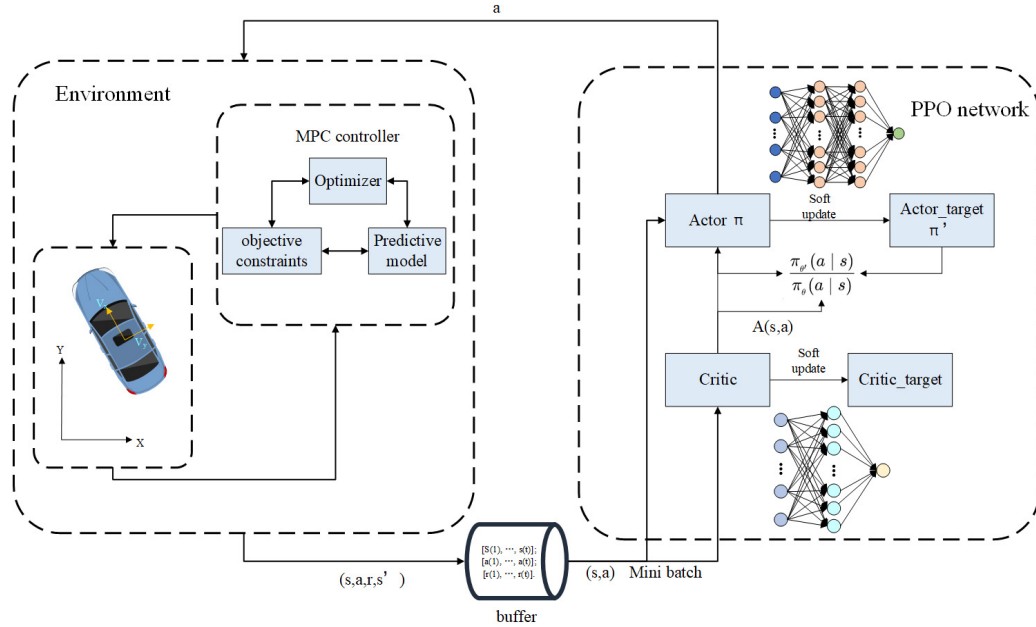

**Figure 3.** PPO-MPC framework.

### 2.3.2. Action and State Space

Considering that the prediction horizon is related to historical trajectory information and road curvature, a state space $S(t) = [c(t), v(t), \delta(t), acc(t), e(t), \cos t(t)]$ was established in this section. $c(t), v(t), \delta(t), acc(t), e(t), \cos t(t)$ represent the curvature of the reference trajectory, velocity, steering angle, acceleration, lateral error, and cost of the MPC

system at time $t$, respectively. Our strategy trains a PPO policy $\pi_\theta^N$ to determine $N_p$ for MPC. At each time step, the system's state is measured, and the policy outputs $N_p$ is used to solve the MPC problem.

The prediction horizon, denoted as $N_p$, is defined as a positive integer representing the maximum value $N_{max}$ of the prediction horizon. In order to adjust the output of the PPO policy, we employ linear scaling. Specifically, we limit the output to $[-1, 1]$, which is associated with the hyperbolic tangent (*tanh*) function, to a new range of 1 to $N_{max}$. This adjustment ensures that the policy output aligns with the requirements of the MPC scheme.

2.3.3. Reward Function

The policy and value function in PPO are learned directly from the reward signal. Thus, an appropriate reward function plays a crucial role in enabling the neural network in PPO to effectively converge towards the optimal solution. Our designed reward function aims to strike a balance between promoting smooth driving and maintaining an acceptable range of tracking deviation. Tracking error is closely related to control performance, and limiting the control output within the constraint range is related to the stability of MPC control. To coordinate MPC and PPO to achieve optimal performance, we design a reward function that takes into account tracking error and control output, and the reward function is denoted as follows:

$$R(t) = w_1 e^{-(\lambda_1|e_1| + \lambda_2|e_2| + \lambda_3|e_3|)} - w_2 H_1 - w_3 H_2 \tag{18}$$

where $e$ is a natural index, $e_1$ is the lateral tracking deviation, $e_2$ is the longitudinal velocity deviation, and $e_3$ is the relative yaw angle error. $H_1$ and $H_2$ are penalty terms. $H_1 = 1$ when either the steering angle or acceleration exceeds the constraint; if both exceed the constraint, $H_1 = 2$, else $H_1 = 0$. $H_2 = 1$ when the lateral tracking deviation is greater than 0.15, else $H_2 = 0$. $\lambda_1$, $\lambda_2$, and $\lambda_3$ are the weights of the tracking error, respectively, and $w_1$, $w_2$, and $w_3$ are the weights of each reward, respectively. Equation (18) adopts the form of an exponential function, which makes the gradient change more drastic, which is beneficial to the training process. The reward increases when the total tracking error decreases, and when errors are 0, the instant reward can be obtained by corresponding points.

**3. Simulation and Training**

In this section, to verify the validity of the strategy we proposed, we conduct training and verification at various speeds using the MATLAB/Simulink simulation platform. In addition, trajectory tracking comparisons are conducted between the PPO-MPC algorithm proposed in this article and MPC with fixed horizons. The vehicle parameters used in the simulation are outlined in Table 1.

**Table 1.** Vehicle parameters.

| Symbol | Description | Value | [Units] |
|:---:|:---:|:---:|:---:|
| $m$ | Vehicle mass | 1600 | [kg] |
| $I_z$ | Yaw inertia of vehicle | 2875 | [kg·m$^2$] |
| $C_f$ | Cornering stiffness of front tire | $12 \times 10^3$ | [N/rad] |
| $C_r$ | Cornering stiffness of rear tire | $11 \times 10^3$ | [N/rad] |
| $l_f$ | Distance from rear axle to center of gravity | 1.4 | m |
| $l_r$ | Distance from rear axle to center of gravity | 1.6 | m |

The PPO algorithm involves a set of hyperparameters that significantly influence the algorithm's performance and training stability. The hyperparameter settings in reinforcement learning must be customized to the specific problem and environment to achieve optimal performance. In this study, the agent collects experiences based on the training set and stops when it reaches a 500-steps experience horizon or the terminal episode. Then it is trained for three epochs using mini-batches of 128 experiences. The objective function

clip factor is set to 0.2 to enhance the stability of training, while the discount factor is set to 0.998 to promote long-term rewards. The Generalized Advantage Estimate (GAE) method reduces the variance in critic output with a GAE factor of 0.95.

Based on the aforementioned simulation environment, training was conducted for up to 10,000 episodes, with each episode spanning up to 500 steps. Figure 4 depicts the training results. The light blue line represents the cumulative reward obtained by the agent at the end of each round; the thick line describes the average reward value of all rounds during the training process. In the early stages of training, PPO reinforcement learning explores various actions through interactions with the environment to achieve the overall optimal outcome. In this experiment, after 300 rounds, the model finally converged stably, showing good training effects.

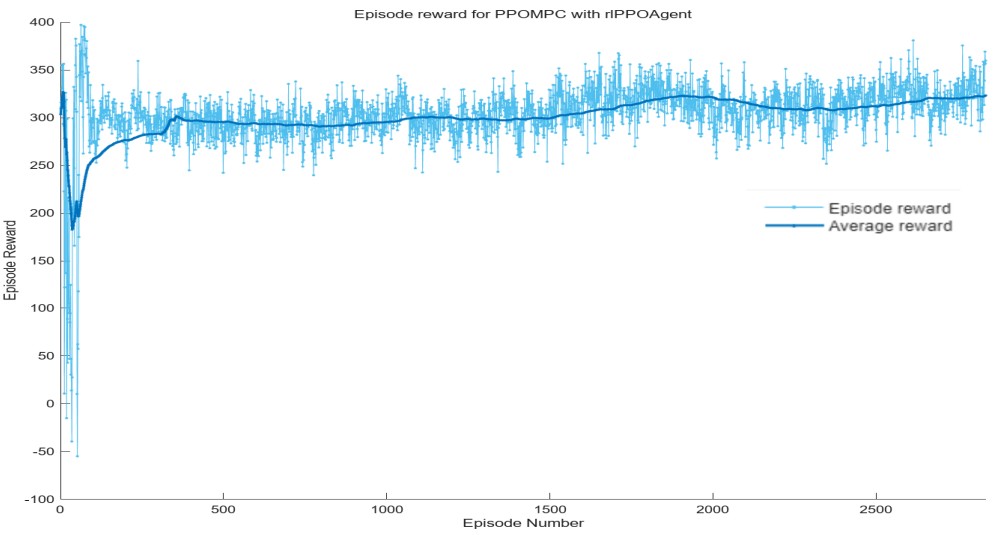

**Figure 4.** The training results.

### 4. Results and Discussion

PPO-MPC is simulated and verified at different speeds, and its trajectory tracking performance is analyzed in this section. In addition, PPO-MPC was compared with MPC under fixed prediction horizons at different speeds, and their performance was discussed. The pre-calculated reference path is shown in Figure 5, with a total length of 12,000 m.

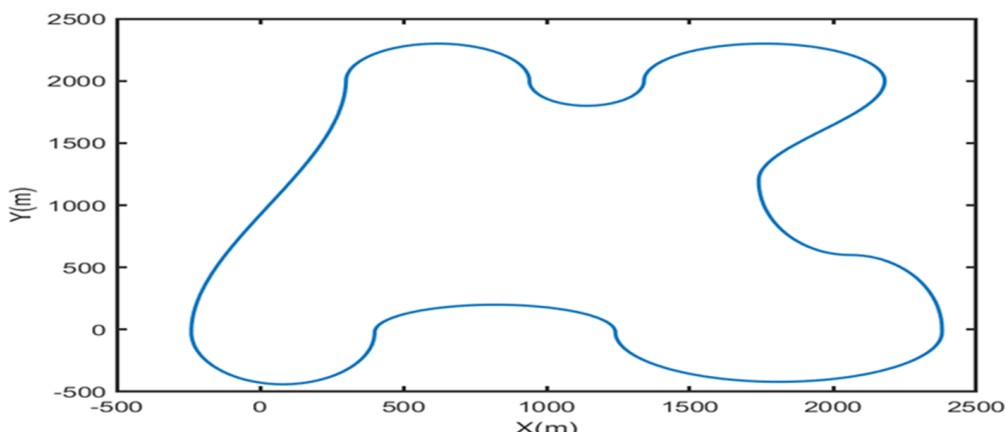

**Figure 5.** Reference path.

### 4.1. Performance of Trajectory Tracking Using PPO-MPC

In this section, simulation verification of PPO-MPC was performed at various velocities ($v = 10$ m/s, $v = 15$ m/s, and $v = 20$ m/s, respectively), and an analysis of its trajectory tracking performance during operation was conducted.

Figures 6–8 illustrate the vehicle control results of our proposed control strategy at $v = 10$ m/s, $v = 15$ m/s, and $v = 20$ m/s. The PPO-MPC controller showed excellent performance by effectively adjusting the acceleration and steering angle output within the predefined constraint range, except for some fluctuations at the beginning of the simulation in Figure 8, which quickly stabilized. Figures 6a, 7a and 8a reveal the vehicle's lateral deviation changes. Large lateral errors may occur where the curvature suddenly changes, and otherwise remain near 0, indicating that the control system has a good ability to keep the vehicle close to the desired trajectory. Figures 6b, 7b and 8b illustrate heading angle error changes, with the maximum value not exceeding 0.05. This shows that the vehicle has good performance in tracking the expected direction. Changes in the yaw rate, as depicted in Figures 6c, 7c and 8c, generally demonstrate low values, suggesting smooth steering operations, mitigated risk of abrupt rolling or sharp turning, and enhanced driving stability. Moreover, Figures 6d, 7d and 8d illustrate speed error changes; even at maximum speed, the speed error for stable operation always remains small, with the error limited within 0.2 m/s.

From the above analysis, it can be concluded that the PPO-MPC controller shows good performance at various speeds. And PPO-MPC effectively maintains the trajectory, direction, and stability of the vehicle, indicating that the strategy has good adaptability in different scenarios.

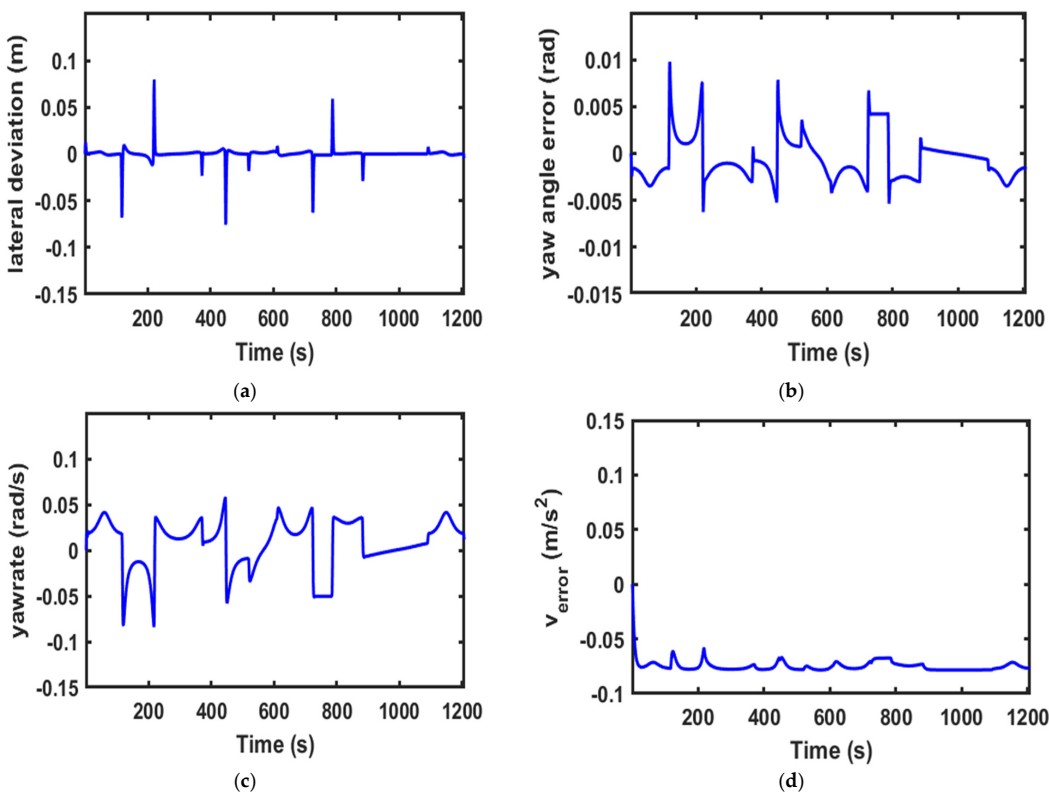

**Figure 6.** Performance of the PPO-MPC controller at $v = 10$ m/s. (**a**) Lateral deviation. (**b**) Yaw angle error. (**c**) Yaw rate. (**d**) Velocity error.

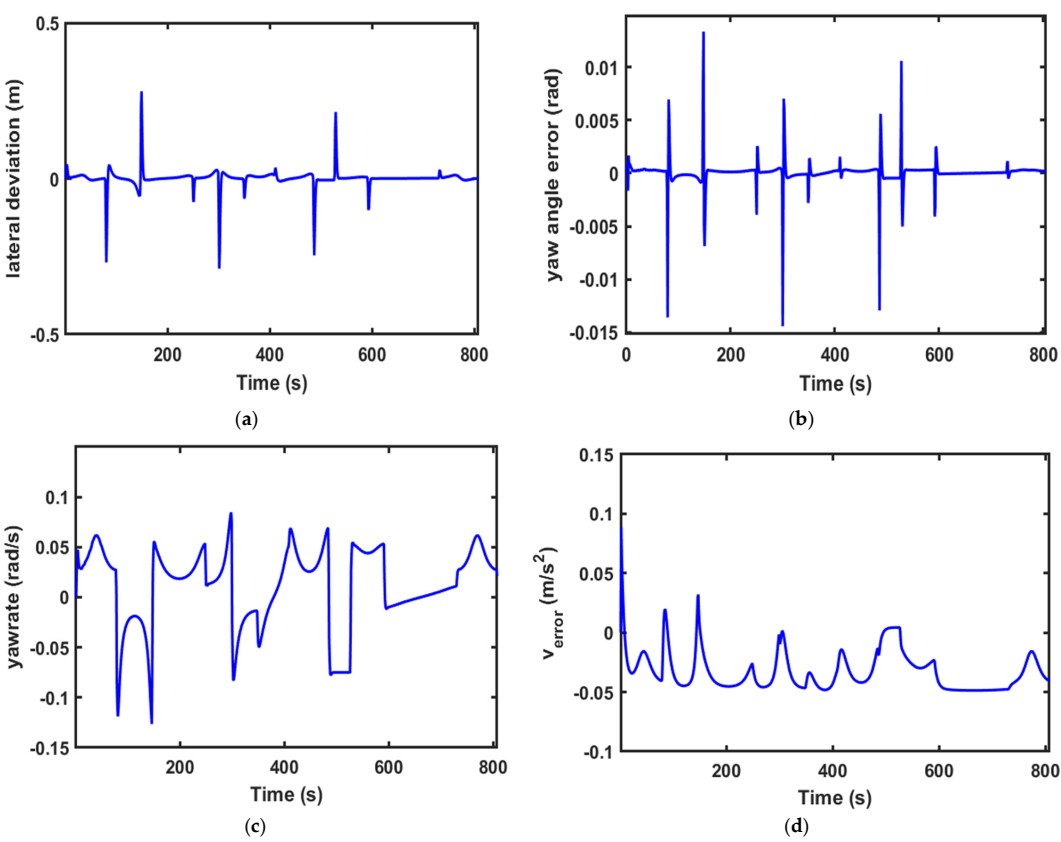

**Figure 7.** Performance of the PPO-MPC controller at $v = 15$ m/s. (**a**) Lateral deviation. (**b**) Yaw angle error. (**c**) Yaw rate. (**d**) Velocity error.

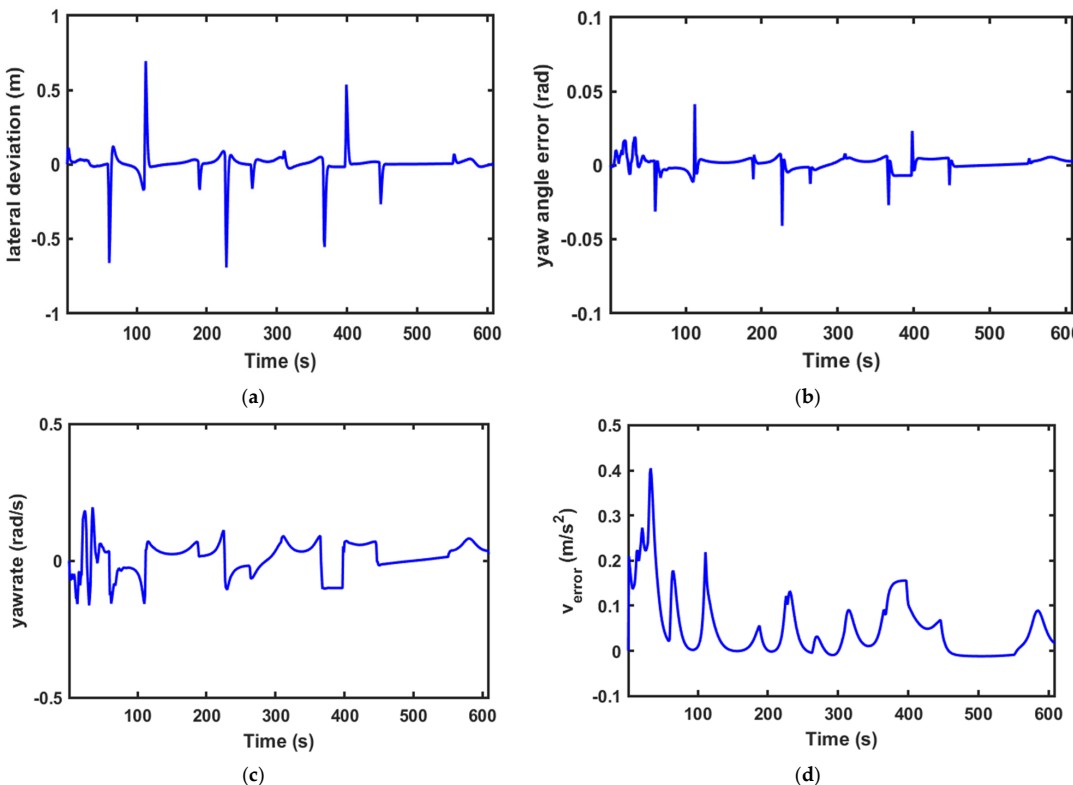

**Figure 8.** Performance of the PPO-MPC controller at $v = 20$ m/s. (**a**) Lateral deviation. (**b**) Yaw angle error. (**c**) Yaw rate. (**d**) Velocity error.

### 4.2. Performance Comparison of the Trajectory Tracking Using PPO-MPC and Model Predictive Control

To further illustrate the advantages of the PPO-MPC strategy, we conducted a comparison between PPO-MPC and the conventional MPC strategy with fixed prediction horizons. In the simulated design, we conducted a comparison between PPO-MPC and MPC with fixed horizons of 10, 20, and 30 and the control horizon set to 3.

Comparisons of the simulation data are shown in Figures 9–11. As shown in Figure 11, it is easy to see that MPC with a fixed range of 10 fails to converge and is unstable at $v = 20$ m/s. Figures 9a, 10a and 11a clearly show that compared with MPC with static prediction horizons, PPO-MPC generally has better lateral deviation and exhibits superior trajectory tracking capabilities. The maximal lateral deviation of MPC may even be twice that of PPO-MPC, suggesting that the MPC with fixed prediction horizons may be subject to have greater lateral disturbance or challenges under certain circumstances. Figures 9b, 10b and 11b show that the heading error of PPO-MPC is almost the same as MPC with static prediction horizons at $v = 10$ m/s, and the PPO-MPC heading error at $v = 15$ m/s is smaller. Moreover, although PPO-MPC jittered at the beginning at $v = 20$ m/s, it quickly stabilized and resulted in a smaller heading error. Figures 9c, 10c and 11c show that, except for the jitter that occurs at the beginning of the simulation when $v = 20$ m/s, the overall performance of the PPO-MPC speed error is smaller.

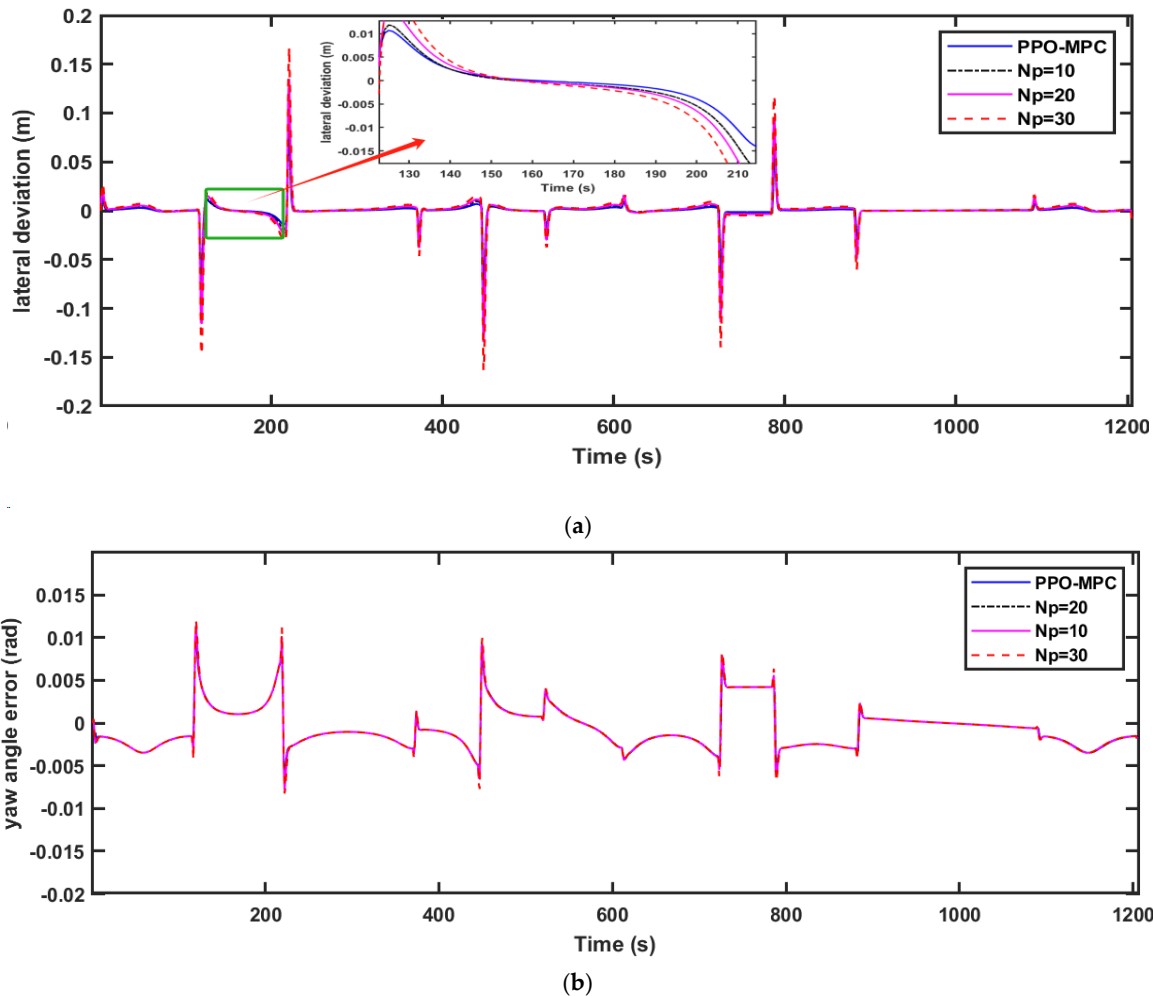

**Figure 9.** *Cont.*

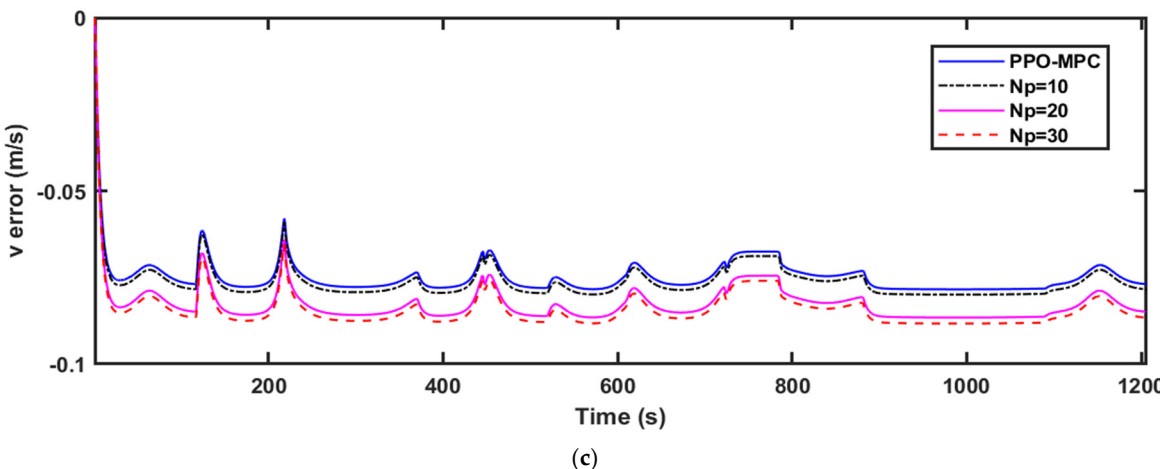

(**c**)

**Figure 9.** Performance comparison of trajectory tracking using PPO-MPC model predictive control at $v = 10$ m/s. (**a**) Lateral deviation comparison. (**b**) Yaw angle error comparison. (**c**) Velocity error comparison.

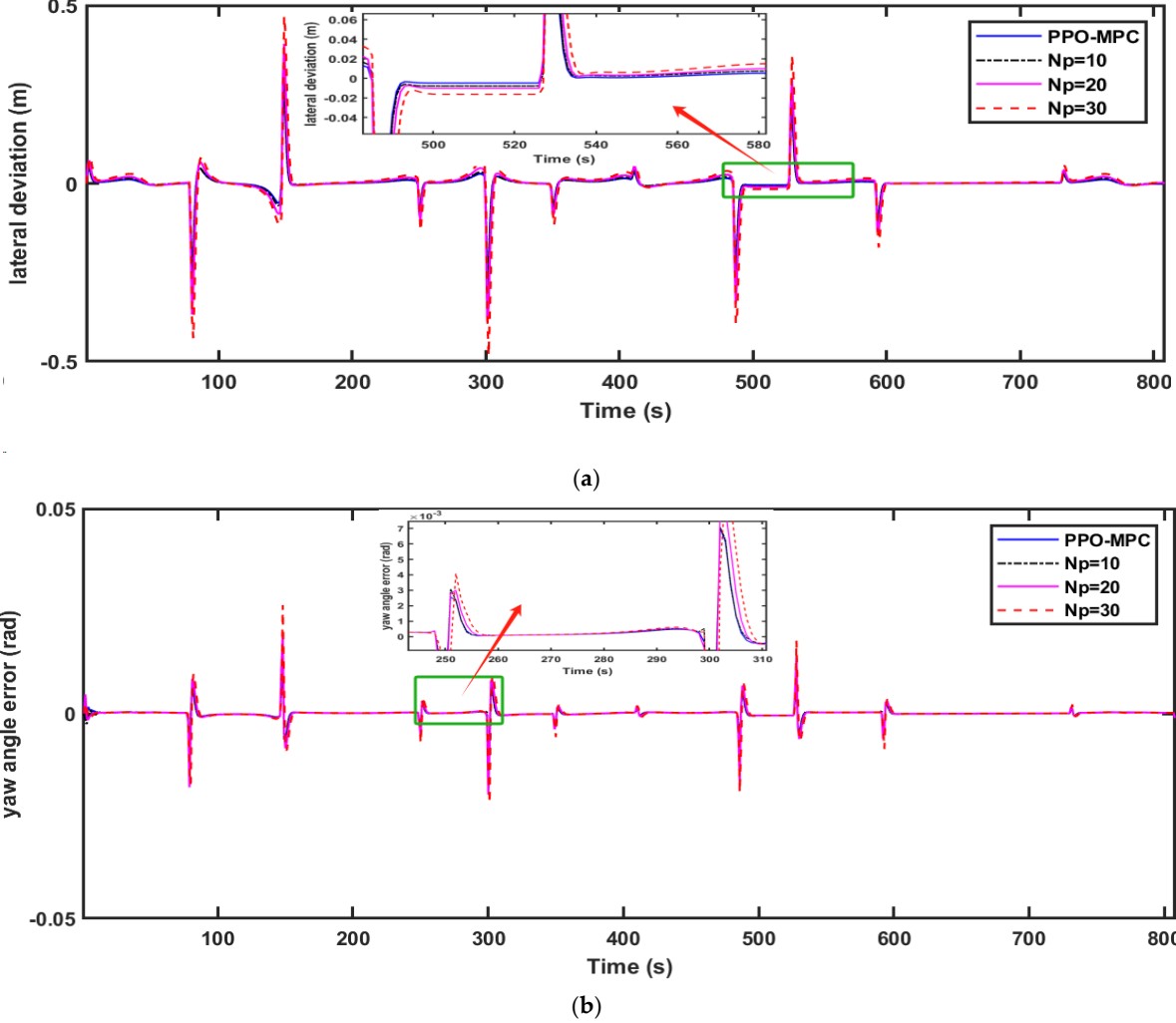

**Figure 10.** *Cont.*

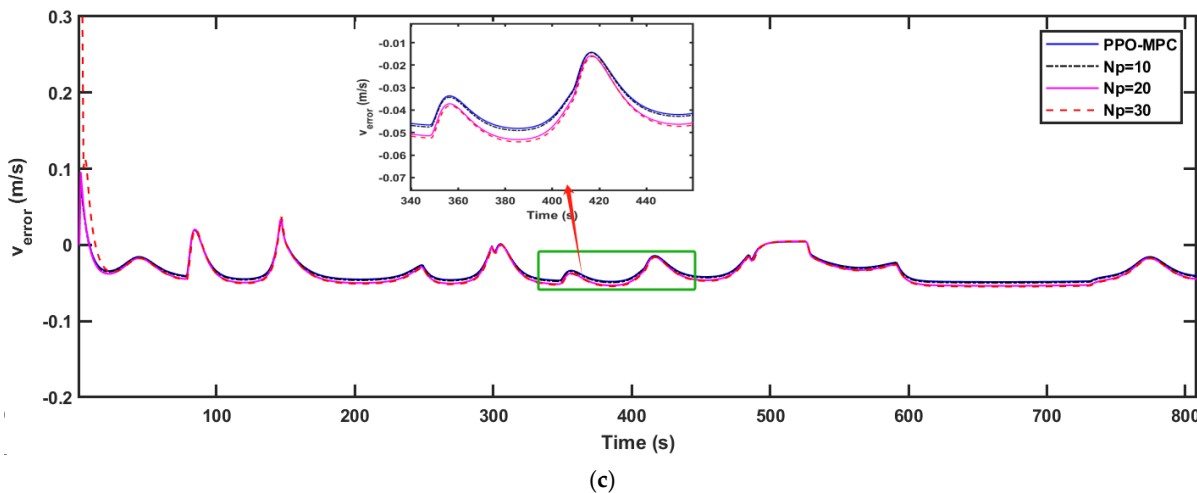

(**c**)

**Figure 10.** Performance comparison of the trajectory tracking using PPO-MPC model predictive control at $v = 15\,\text{m/s}$. (**a**) Lateral deviation comparison. (**b**) Yaw angle error comparison. (**c**) Velocity error comparison.

(**a**)

(**b**)

**Figure 11.** *Cont*.

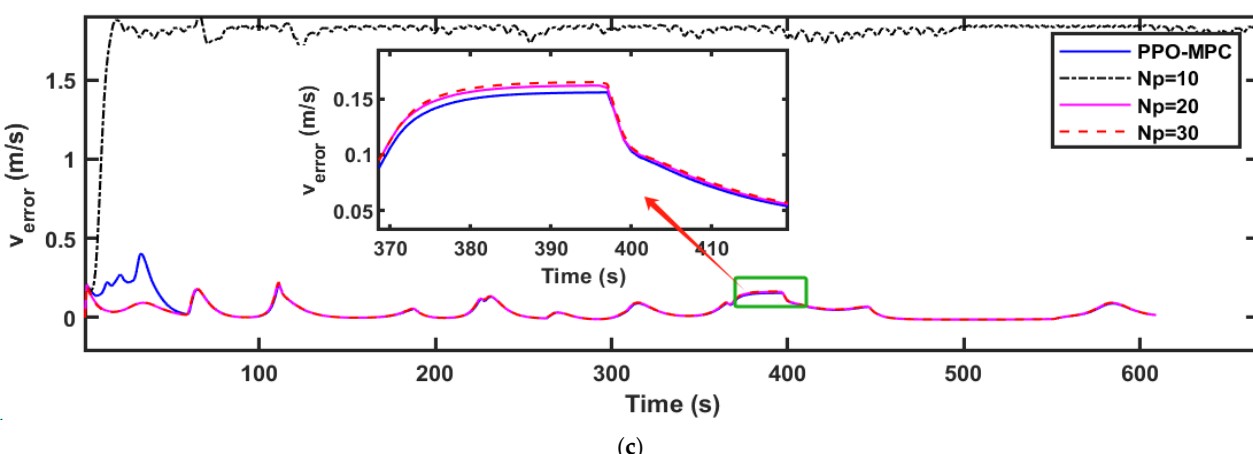

**(c)**

**Figure 11.** Performance comparison of the trajectory tracking using PPO-MPC model predictive control at $v = 20$ m/s. (**a**) Lateral deviation comparison. (**b**) Yaw angle error comparison. (**c**) Velocity error comparison.

According to [37], we introduce an index which quantitatively measures the tracking performance and is achieved as

$$Q_{track\_i} = \sqrt{\frac{\sum\limits_{j=1}^{T/T_s} \left( y_{ref}(j) - y(j) \right)^2}{T/T_s - 1}} \tag{19}$$

where $T$ represents the simulation duration time and $T_s$ represents the controller sampling step. The tracking performance indexes of simulations are provided in Table 2.

**Table 2.** Tracking accuracy indexes of simulations.

| Velocity | $v$=10 m/s | | | $v$=15 m/s | | | $v$=20 m/s | | |
|---|---|---|---|---|---|---|---|---|---|
| Index | $Q_{tr\_lat}$ | $Q_{tr\_yaw}$ | $Q_{tr\_v}$ | $Q_{tr\_lat}$ | $Q_{tr\_yaw}$ | $Q_{tr\_v}$ | $Q_{tr\_lat}$ | $Q_{tr\_yaw}$ | $Q_{tr\_v}$ |
| PPO-MPC | 0.0022 | 0.0019 | 0.0749 | 0.0107 | 0.0004 | 0.0339 | 0.0343 | 0.0035 | 0.0536 |
| Np = 10 | 0.0036 | 0.0019 | 0.0763 | 0.0119 | 0.0005 | 0.0345 | 0.0814 | 0.0585 | 1.7936 |
| Np = 20 | 0.0043 | 0.0020 | 0.0826 | 0.0163 | 0.0006 | 0.0374 | 0.0415 | 0.0036 | 0.0436 |
| Np = 30 | 0.0057 | 0.0020 | 0.0843 | 0.0221 | 0.0006 | 0.0474 | 0.0443 | 0.0035 | 0.0443 |

$Q_{tr\_lat}$, $Q_{tr\_yaw}$, and $Q_{tr\_v}$ denote the lateral, heading, and speed tracking accuracy, respectively. While ensuring dynamic stability, it is not difficult to find that, compared with MPC, the PPO-MPC tracking accuracy is improved, except that the speed tracking accuracy of PPO-MPC at $v = 20$ m/s is smaller than MPC, with $N_p = 30$. Analysis shows the superiority of the proposed PPO-MPC path-tracking controller.

**5. Conclusions**

In this paper, we present a novel PPO-MPC strategy, which integrates proximal policy optimization (PPO) with model predictive control (MPC), using the PPO reinforcement learning algorithm to dynamically adapt the prediction horizon of MPC. The proposed strategy was evaluated and validated using the MATLAB/Simulink simulation environment across three distinct operating speeds. Additionally, comparisons were conducted against conventional MPC employing static prediction horizons under analogous conditions. From the analysis of simulation results, it can be seen that the PPO-MPC framework is better than the traditional model predictive controller with a fixed prediction range, and has superior tracking performance and robustness.

In the future, PPO-MPC can be explored through multi-objective optimization methods. Co-optimizing the prediction horizon with other key MPC parameters (such as control weights and constraints) is expected to achieve more powerful and efficient vehicle control strategies. And it may achieve adaptive trajectory control in various scenarios. In addition, adding the RL differential prediction model to the MPC prediction model to achieve better adaptive control is also a major idea for improving automatic driving control performance.

**Author Contributions:** Z.C.: design of methodology and writing; J.L.: conducted train and validation simulation and wrote the manuscript; P.L.: data analysis and reviewed the manuscript; O.I.A.: edited and reviewed the manuscript; Y.Z.: wrote the manuscript. All authors have read and agreed to the published version of the manuscript.

**Funding:** Open access funding provided by the Hainan Province Key R&D Plan Project, China (No. ZDYF2024GXJS020, No. ZDYF2021GXJS002) and Hainan Natural Science Foundation, China (No. 523RC441).

**Data Availability Statement:** Data are contained within the article.

**Conflicts of Interest:** The authors declare no competing interests.

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
