# Peer review of "Prediction Horizon-Varying Model Predictive Control (MPC) for Autonomous Vehicle Control"

_electronics, doi:10.3390/electronics13081442_

Round 1

Reviewer 1 Report

Comments and Suggestions for Authors

This paper introduces a Reinforcement Learning (RL) method into the Model Predictive Control (MPC) framework for improving a vehicle’s trajectory tracking performances. According to it, the contribution lies in that the RL module can automatically change the MPC’s prediction horizon so that the vehicle can better adjust to the various complex road scenarios. As a reviewer, there are some problems for this paper to be accepted for publiation.  My detailed comments are given below.

------ Contribution of this work ------

I think the contribution of this paper is weak. The modelling of a vehicle’s dynamics and RL algorithm, PPO, are standard approaches. Hence, the key is to design a reward function for the training RL agent to dynamically tune the prediction horizon of MPC.

1> The reward factors are not good measurements of the influence of MPC's time horizon.

2> Another issue is related to the figure showing the training results.

------ Experiments ------

1> Experiment 1 (Section 4.1):  PPO-MPC is tested for trajectory tracking with 3 different velocities. The aim is to show the tracking performance of this method and it does demonstrate that PPO-MPC can perform well given settings. But  I think the analysis of these 3 cases can be combined. No need to repeatedly explain the same thing over and over again.

2> Experiment 2 (Section 4.2): PPO-MPC is compared with fixed time horizon MPC, but it is only compared with one time horizon . Comparison with other time step settings should also be considered.

3> Since RL is used for choosing time horizon, some statistics on how time-horizon varies and other performance measurements (such as cost improvement) are desired.

------ Other Issues ------

This paper’s literature review needs to be strengthened. Please pay attention to some recent studies, e.g., Autonomous Vehicle Platoons in Urban Road Networks: A Joint Distributed Reinforcement Learning and Model Predictive Control Approach; Model Controlled Prediction: A Reciprocal Alternative of Model Predictive Control; Moment-Based Model Predictive Control of Autonomous Systems; and Deep Learning-Based Model Predictive Control for Continuous Stirred-Tank Reactor System.

This paper has some formatting mistakes, such as,

1> all the equations should be cited with the given index number instead of being repeatedly written in the text.

2> from line 215 to 223, the sentences are quite messy to read and some reference is missed at line 306.

3> Figure 4 is not clear in showing the training results. Some explanation is needed for what the yellow and blue lines stand for. A proofreading is truly needed for the whole article. 

Comments on the Quality of English Language

See above

Author Response

All responses to the revision comments are included in the uploaded file.

Reviewer 2 Report

Comments and Suggestions for Authors

This paper proposes an autonomous vehicle control strategy using the Proximal Policy Optimization (PPO) algorithm to adjust the prediction horizon, enabling MPC to achieve optimal performance. The proposed model is named PPO-MPC.

The paper is well-written and well-structured. The model seems to be very well described. The reference list should be refreshed with the latest related work.

The paper has several issues that should be addressed before publishing:

1. It seems that there are not so small number of papers dealing with PPO-MPC vehicle control. Authors should explain how their contribution differs from the previous contributions.

2. Fig. 1 is not described with all elements.

3.  The text describing Fig 6 should contain a reference to figure subplots, e.g. Fig. 6 a). Whole Fig 6 should be better described in the text.

4. Authors should try to precisely quantify the better performance of PPO-MPC compared to MPC.

5. The Conclusion section is too short, it should be expanded. The authors should summarize the results in the conclusion section and describe possible further work directions.

Author Response

(The authors gave the same response as above.)

Round 2

Reviewer 2 Report

Comments and Suggestions for Authors

The authors answered all addressed comments.